# Biodegradable Nonwovens with Poultry Feather Addition as a Method for Recycling and Waste Management

**DOI:** 10.3390/polym14122370

**Published:** 2022-06-11

**Authors:** Jagoda Jóźwik-Pruska, Krystyna Wrześniewska-Tosik, Tomasz Mik, Ewa Wesołowska, Tomasz Kowalewski, Michalina Pałczyńska, Damian Walisiak, Magdalena Szalczyńska

**Affiliations:** Łódź Institute of Technology, Łukasiewicz Research Network, 90-570 Lodz, Poland; krystyna.wrzesniewska-tosik@lit.lukasiewicz.gov.pl (K.W.-T.); tomasz.mik@lit.lukasiewicz.gov.pl (T.M.); ewa.wesolowska@lit.lukasiewicz.gov.pl (E.W.); tomasz.kowalewski@lit.lukasiewicz.gov.pl (T.K.); michalina.palczynska@lit.lukasiewicz.gov.pl (M.P.); damian.walisiak@lit.lukasiewicz.gov.pl (D.W.); magdalena.szalczynska@lit.lukasiewicz.gov.pl (M.S.)

**Keywords:** biodegradation, keratin, feather, poultry waste, nonwovens

## Abstract

Geotextiles are used for separation, drainage, filtration and anti-erosion protection sealing, as well as to improve plant vegetation conditions. The research objective of this study was to verify the influence of the addition of poultry feathers on accelerating the biodegradation of nonwovens in cultivated soil. The tests were carried out in laboratory conditions and were based on the assessment of weight loss. The experiments confirmed the positive effects of the presence of waste that was rich in keratin on the time required for the biodegradation of the tested materials (the period of biodegradation was 8–24 weeks). Additionally, the influence of the biodegradation of the tested materials on the ecotoxicity was investigated and showed no negative effects on the microbiological activity (10^6^ cfu). The research also included the determination of the carbon to nitrogen ratio of the test medium (blank, 12–14:1; with feather addition, 19–20:1). A statistical analysis revealed a correlation between the mechanical properties and the period of biological decomposition. This research was an important step for the management of poultry feather waste in agricultural applications. The tested materials could be seen an alternative that meets all ecological criteria, which seems to be a golden solution that not only allows the delivery of important nutrients to the soil, but also manages waste in an environmentally safe manner.

## 1. Introduction

According to the Organisation for Economic Cooperation and Development (OECD), “biodegradation” is the process of the decomposition of organic substances by microorganisms into simpler substances, such as carbon dioxide, water and ammonia [1,2]. The microorganisms require energy and oxygen, carbon, phosphorous, sulphur, nitrogen, calcium, magnesium and other elements to grow and reproduce. Organic substances are oxidised into carbon dioxide and water through an exothermic process and the obtained energy is partially used by the microorganisms and the rest is lost as heat. The process can be especially observed in composting. Biodegradation can be conducted both under aerobic and anaerobic conditions [3]. Bacteria, fungi, insects, worms and many more organisms participate in the breakdown of various materials. Biodegradation is essential for nature and the whole ecosystem because it provides the opportunity to decrease waste and produce nutrients that are crucial for the growth of new life [4,5]. Recently, biodegradability has become a requirement for materials that are used in everyday life and is one of the essential features for evaluating their sustainability [4].

A wide range of tests can be applied to examine the biodegradability of a product. The choice of method refers to the type and properties of each sample. The biodegradation process is conducted mainly in water, soil and compost environments [6]. The estimation of biodegradability is mainly based on a calculation of weight loss or by an evaluation of the production of CO_2_. Each method has its own advantages and disadvantages. Estimation on the basis of released CO_2_ only relates to the measured and calculated theoretical carbon content of the sample. In the method that is based on the weight loss calculation, there is the risk of losing microscopic parts of the sample. Nevertheless, in our opinion, the second method provides a more complete picture of the degradation process and the behaviour of the sample over time (influence on the structure, surface, disintegration ability, etc.).

Chemical fibres are manufactured for various purposes, including textiles and agriculture. The worldwide production of fibres is growing every year. The literature reports that the global production output of chemical fibre (organic and synthetic) industry has reached 80.9 million metric tons [7]. Synthetic polymers are the most commonly used materials to produce manmade fibres. Due to the fact that they are not biodegradable, their application needs to be limited and products should be reused to a limited extent. The resistance of these materials to biological breakdown is crucial because of environmental pollution. Their build-up in the environment results in the release of toxic pollutants, which then influence living organisms within the soil and water [4].

The importance of fibre-reinforced composites in the manufacture of a wide range of industrial products is still increasing. Special interest is focused on the replacement (full or partial) of synthetic polymers with biopolymers, including keratin. This polymer can be obtained from sheep wool, poultry feathers, horn, nails and many other sources. Its chemical properties allow for the use of keratin as a thermoset material, which can be linked to other polymers [8,9,10,11]. Biodegradability, biocompatibility and fire-retardant capability are among the valuable properties of keratin [8].

It should be highlighted that according to the regulations of the European Parliament [12], the promotion of harmonious and sustainable economic growth should be carried out with respect for the natural environment. The modern approach to environmental protection enforces the creation and implementation of new technologies, especially those that contribute to the elimination of pollution at the source.

In a leading company on the Polish market, the amount of waste in the form of feathers is 6 tons per day, which is processed into industrial flour. The European Union (EU) has banned its use in fodder, which has caused a problem with the legitimacy of processing feathers that are a by-product of slaughter into industrial meal.

The processing of keratin-rich feathers allows us to obtain diverse thermoplastic biocomposites and translates into improvements in economic returns for the poultry industry. The composition of feather keratin is based on small protein molecules (molecular weights of 10–30 kDa). Extensive internal bonding results in thermal and mechanical stability [13]. Undeniably, its low cost and natural abundance makes feather keratin a valuable material for the production of biodegradable polymers for various applications [8].

The literature has reported the incorporation of feather keratin in various synthetic polymers, such as polypropylene (PP) and polyethylene (PE) [14,15]. Cheng et al. [16] investigated the possibility of incorporating feather fibres into polylactic acid (PLA). Other researchers combined feathers with polyurethane [11,17].

The biodegradation of fibres starts with changes in their structures or compositions. Chemical changes can be examined by the application of Fourier transform infrared spectroscopy (FTIR) or infrared spectroscopy. Degradation can be also estimated by visual observations and microscopy [4,18]. According to the European Standard EN 14995 Plastics—Evaluation of Compostability—Test scheme and specifications (2009) [19], the biodegradation process of a sample cannot exceed 24 weeks. The progress of decomposition is estimated and calculated on the basis of weight loss. While the biodegradability of various textiles is a desirable feature, sometimes the final product design requires sufficient resistance to degradation in order to provide long-term use [4]. Yet, the safety of use should be examined.

Poultry feathers are a by-product of animal origin, which are obtained during poultry slaughter. There are about 33 plants and 100 slaughterhouses operating within the poultry industry in Poland, of which Cedrob S.A. is Poland’s largest. Cedrob S.A. belongs to the Cedrob Group, which also includes Gobarto S.A. (the leading pig meat producer) and Cedrob Passau GmbH (the Cedrob Group’s representative on the German market).

The further processing of feathers into poultry meal is realised by Utilisation Plants, which was used as a protein additive for industrial feeds until 2004. After the introduction of the Regulation of the European Parliament and Council (EC) No. 999/2001, the ban on its use in livestock nutrition has caused a problem with the reasonableness of processing of feathers, among other materials, from slaughter into industrial meal. The only other uses of such meal are possibly as organic fertiliser for field fertilisation or as an addition to incineration in industrial boiler rooms. Both applications have no economic or environmental justification and can cause environmental and health hazards.

In response to the need to manage feather waste, the Team of Keratin Composites from the Łukasiewicz Research Network, Institute of Biopolymers and Chemical Fibres, aimed to develop innovative feather-based nonwovens that are characterised by additional functionalities and advantages, which are derived from the use of feather keratin, such as tailor-made biodegradation that is adjusted to the crop duration, the input of organic nitrogen into the soil, zero waste at their end of life and cost-competitive materials. The nonwovens that are obtained by the needle punching method consist of wool and feather-based keratin fibres and can be used for agricultural applications. This method has now been patented (P.430284 (19 June 2019) “Method for producing fluffy composite nonwoven fabric”). In this context, the main objective of this study was to create a concept for a waste management method for the by-products of animal production that is desirable from the point of view of economics and social effects through the exploitation of underutilised waste, in order to obtain added value raw materials for the agricultural sector, such as feather-based nonwovens. The technology of the designed solution assumes that the share of waste material in the form of feathers is at a level of about 50%.

There are many reasons why developed nonwovens are suitable for agricultural use:They are safe for the environment and human health by ensuring a reduction in biomass waste, in the form of feathers, that is deposited and pollutes the environment;The developed innovative nonwovens are made of biodegradable raw materials of natural origin;We have the ability to control the time of microbial decomposition by adjusting the share of feather fractions in the nonwovens;They have high efficiency with low financial outlay (i.e., the market price of the developed products is much lower than that of fossil-based products due to the fact that they are made from waste materials);Nonwovens that are made from natural waste resources can be used soil improvement agents because they contain significant amounts of fertilising ingredients within their structure, which can then be used to meet the nutritional needs of crops.

The paper presents the results of the biodegradation of keratin-based nonwoven fabrics in a soil environment and the influence of their mechanical properties on the process. The influence of the composition (feather amount) on the susceptibility of the material to biodegradation was examined and a statistical analysis was performed.

## 2. Materials and Methods

This study was carried out on two groups of protective nonwovens with the addition of keratin fibres in the form of feathers from a poultry slaughterhouse (i.e., Cedrob S.A., Ciechanów, Poland, which is Poland’s largest poultry producer).

The technology of the designed solution assumed the share of waste material, in the form of feathers, in the nonwovens to be at the level of about 50%. The percentage content of feathers in the nonwovens was estimated on the basis of a weight study: the difference in weight of the nonwoven containing the feathers and that of the reference nonwoven (without feathers). The first group was composed of Trevira bico “type 256” and the second was of Trevira bico “type 453”. Additionally, the reference samples, without feathers, were examined and were marked as “0”. The nonwovens were designed and intended to cover the grassy-bean mixture on difficult terrains (new dumps, heaps, railway embankments, ski slopes, etc.), on which obtaining a good sodding is very difficult. Table 1 presents the compositions of the samples that were tested. Table 2 presents selected mechanical properties of the materials. The SEM photo-documentation can be found in the Appendix A.

The biodegradation tests were carried out at the laboratory scale. For the experiments, samples taken from nonwovens that were produced at a quarter-technical scale were used. Each 5 × 5 cm sample was tested in triplicate under the conditions of repeatability and reproducibility. For each final result, the components of the uncertainty of the measurement were determined. The method used has been validated. Cotton (100%) was used as reference material. According to the available standards, biodegradable materials should achieve 90% decomposition within a maximum period of 24 weeks.

The biodegradability tests were conducted in soil under the controlled conditions of temperature (30 ± 2 °C) and humidity (60–75%). The start of each test was preceded by an examination of the microbiological activity of the medium (soil) in order to ensure appropriate conditions (≥10^6^ cfu). The samples were placed in research reactors that were filled with the test soil and then stored in a heat chamber, which enabled the control and maintenance of the set environmental parameters (temperature and humidity). The incubation process was carried out at a constant temperature for a maximum period of 24 weeks with the daily humidity control of the test medium. Within the designated periods, the progress of the biodegradation process in aerobic conditions was controlled. Photo-documentation was also obtained. Additionally, ecotoxicity tests were conducted and showed the influence of the decomposition of the developed nonwovens on the microbiological activity of the microorganisms inhabiting the soil. This research was carried out in accordance with the accredited research procedure of the “Assessment of the influence of natural and synthetic materials on soil microflora”, which was developed on the basis of the relevant international standards (EN ISO 7218:2008; EN ISO 11133; EN ISO 11133:2014-07/A1; EN ISO 4833-1:2013-12; EN ISO 19036:2020-04).

The mechanical properties were also tested according to the relevant international standards (PN-EN ISO 9073-2:2002; PN-EN 29073-1:1994; PN-EN 29073-3:1994; PN-EN ISO 9073-4:2002).

Due to the complexity and heterogeneity of the research materials, it was not possible to describe the reaction stoichiometry in detail. Nevertheless, the research included the determination of the carbon to nitrogen ratio within the test medium during the biodegradation process. The aim was to control the C:N ratio and verify that it was not negatively influenced by the sample decomposition.

The data were then statistically evaluated using a statistical analysis package (StatSoft, Poland STATISTICA, version 9.0.). The Shapiro–Wilk test was used to check for the normal distribution of the results. When the results were non-parametric, the Mann–Whitney U test was used to determine any differences between the results in both groups. The level of statistical significance was defined as *p* < 0.05. A correlation analysis was also performed.

## 3. Results

The mechanical properties of the studied materials were tested. Table 2 presents the obtained results.

The biodegradation degree (mass loss) was calculated for the tested groups of nonwovens. The level of mass loss varied considerably between these groups. After 24 weeks, the level of biodegradation in Group I reached an average of 89.6% ± 2.67, while in the Group II, all samples reached 100% within 8–24 weeks. Table 3 presents the photo-documentation of the progress of the process for selected samples. The application of the Shapiro–Wilk test showed that the hypothesis regarding the data being normally distributed could be rejected (*p* < 0.05). The differences between the levels of mass loss of the two groups were found after performing a Mann–Whitney *U* test. Considering a difference in the *p*-value of < 0.05 to be statistically significant, the compositions of the nonwovens had an influence on their biodegradability.

For the samples from Group II, a detailed analysis of the effects of the addition of poultry feathers on biodegradation was performed. Figure 1 presents the ratio of the biodegradation times of the materials.

In order to present the differences between the selected features of the tested samples in Group II, an ANOVA test was performed (Table 4).

A correlation analysis was also performed in order to present the interdependencies between the selected properties of the materials. Correlations between the mechanical properties and the degree of the mass loss/time of biodegradation were checked (*p* < 0.05) and the obtained results are presented in Table 5.

In order to illustrate the relationships between the examined features, a cluster analysis was performed for the samples in Group II (Figure 2). The test organised items (features) into groups, or clusters, on the basis of how closely associated they were.

The determination of the carbon to nitrogen ratio in the test medium was determined during the biodegradation process. The obtained results showed differences between the blank sample of soil and the samples of soil during/after the decomposition of samples. While the C:N ratio for the blank sample of soil was constant throughout the trial (12–14:1), the ratios were higher for the media in which the samples were buried (19–20:1), especially during the first weeks of the trial.

The main aim of the ecotoxicity tests was to investigate the influence of the nonwovens, with and without feathers addition, on the microbiological activity of the tested substrate (soil). The conducted tests showed no toxic effects on the microorganisms. This testing was very important due to the essential role of microorganisms in the biodegradation process.

## 4. Discussion

The literature reports that chicken feathers have unique properties. The barb is a protein fibre that has high flexibility, low density and a good spinning length. The rachis has low rigidity and low density. These features make chicken feather barbs a good composite for manufacturing textile products, either on their own or in structural interactions with other fibres [20].

Natural fibres are divided into three main groups, based on their origin: plant (cellulose) fibres, animal (protein) fibres and mineral fibres. Their compositions (cellulose/protein content) influence their mechanical properties and the biodegradation process [21,22,23]. The main medium for the decomposition of polymer waste is soil, which is characterised by varied biodiversity [24]. The time required for the biodegradation of bioplastics depends on the substrate properties [22].

Recently, interest in geotextiles within environmental engineering has been increasing. Both synthetic and biodegradable materials are used. Separation, drainage, filtration, anti-erosion protection sealing and improvements in plant vegetation conditions are among their most important functions. Biopolymers and natural fibres could replace synthetic materials in up to 50% of applications [22]. This is very promising, especially as synthetic fibres are usually not subject to biological degradation. It should be highlighted that due to the growth in consumer and industrial demand for environmentally friendly products, the use of raw materials that are obtained from natural sources has increased significantly. The replacement of synthetic materials, such as PP and polyester, with natural biopolymers (e.g., poly(lactic)acid) is essential because of the amount of waste that is produced due to consumption. According to the US Environmental Protection Agency, 14.3 million tons of textiles were discarded in 2012. It should be pointed out that only 15.7% of this waste was recovered [25]. Recently, geotextiles that are made from synthetic fibres have been considered more critically. A great emphasis is now placed on the application of natural fibres, or biofibres, from renewable sources [26]. Chicken feather-based geotextile materials seem to be a promising solution within agriculture due to the properties of keratin. The feather fibre can preserve soil, increase moisture content and decrease the compaction of soil [26,27].

The study described in this paper showed the great potential of nonwovens that are made with the addition of feathers, especially in that they are biodegradable. The use of feather waste in the production of nonwovens as agricultural products could be the perfect solution for the management of hazardous waste while simultaneously enriching the natural environment. The literature reports [28] that chicken feather fibres can be used as a cheap raw material for nonwoven production. Yet, due to poor length, they need to be combined with other material during the production process.

The main purpose of the biodegradation tests was to demonstrate the impact of the material on the environment. Due to the huge problems with the management of post-consumer waste, the ability to biodegrade is a desirable feature for materials. The process also provides nutrients that are crucial for the growth of new life.

The literature reports that the rate of biodegradability of various materials depends on the nature of the polymer and the structure of the fibres [29]. Our obtained results showed no influence of the presence of feather addition on the biodegradation process in Group I (correlation coefficient = 0.24). Different results were obtained for Group II (correlation coefficient = −0.46; *p* < 0.05). Here, it could be seen that in most cases, the addition of feathers shortened the degradation time of the nonwoven fabrics. It could be assumed that the chemical compounds that are present in feathers (mainly keratin) had a positive effect on the activity of the soil microorganisms, thereby accelerating the biodegradation time of the whole sample.

It should be highlighted that animal remains that are rich in α-keratin are relatively quickly biodegraded by keratinolytic microorganisms, which use native keratin as a source of C, N, S and energy. The mechanisms of degradation are not fully known [27,29,30,31]. One of the initial theories was presented by Raubitschek [32], but this was discarded after the discovery of keratinase [33,34]. Other theories pointed to, inter alia, enzymatic keratin digestion by keratinolytic enzymes and the sulphuric amino acid metabolism of microorganisms as the basis of decomposition [35,36,37].

A wide range of bacteria, actinomycetes and filamentous fungi have been characterised as keratinolytic microorganisms. Special attention should be paid to bacteria belonging to the genus Bacillus, such as *B. subtilis*, *B. pumilus*, *B cereus*, *B. coagulans*, B. *licheniformis* or *B. megatherium* [28,35]. The type of microorganism is not the only factor that is essential for the degradation of materials. Their activity and the properties of the material itself (ecotoxic effects) are also extremely important. The conducted ecotoxicity tests showed no negative influence of the decomposition of the materials on microbial activity and allowed us to establish the ecological characteristics of the tested materials.

Tesfaye et al. [20], on the basis of mechanical properties, assumed that feathers could be applied in geotextiles and road construction applications, as well as the textile industry, energy industry (as insulation materials) and packaging industry. Agriculture application is of special interest because of the water-holding capacity of the feather fibres, which could improve the moisture content of soil. According to scientific reports, feather composting is a safe, sanitary and cost-effective technology that could allow us to obtain products (compost) that could then be used as fertilisers. Their decomposition in soil environments results in an increase in carbon (C), nitrate nitrogen (N-NO3) and sulphur sulphate (S-SO4) concentrations, which are easily absorbed by plants [5,36,37,38]. Our study was in accordance with the literature data (C:N ratio).

The mechanical properties of materials influence the times of biodegradation. It could be concluded that higher values of tensile strength and tear resistance reflected the ability of microorganisms to resist the decomposition of the nonwoven fabrics. Simultaneously, it should be pointed out that higher amounts of feather addition shortened the biodegradation times. The nonwovens are supposed to protect seeds, prevent soil from washing off, limit the impact of rainfall on soil aggregates and inhibit rainwater runoff. The tensile strength of the nonwovens could be a deciding factor in its long-term durability and service life. Hence, good tensile strength is a necessary parameter for them. The base weights of the nonwovens are also important. A heavier material would press the seeds into the soil, cover them and improve conditions for seedling germination and development.

The cluster analysis showed that there was a strong linkage between the mechanical properties. This group of features was associated with the number of weeks of biodegradation. In turn, this cluster was connected to the feather amount.

Jin et al. [39] tested nonwoven fabrics (PE/PP) with the addition of duck feather fibres. Their study revealed the good mechanical properties and sorption capacity of the bicomponent, indicating its potential application as a material for textile dyeing effluent treatment. Soekoco et al. [40] applied chicken feather waste to the production of nonwoven insulator material. The obtained material, which was based on PP, showed higher tensile strength values than commercial insulator material.

There is little evidence in the literature regarding the biodegradability of nonwovens with the addition of feather waste. Mrajji et al. [41] investigated the effects of nonwoven structures on the mechanical, thermal and biodegradability properties of feather-based nonwoven materials that were reinforced by polyester composites. The obtained results showed that the introduction of feather waste into the matrix slightly reduced the degradation process time. This confirmed that the type of polymer and the structure of the fibre have a great influence on the rate of biodegradation.

Interest in the implementation of feather waste in the production of nonwovens is still increasing. Casadesús et al. [42] proposed a method for the management of feather waste in the production of sound-absorbing nonwoven materials. The authors investigated the environmental impacts of the solution using life cycle analysis (LCA) methodology and the utility aspect. The obtained results showed that higher amounts of feathers lowered the environmental impacts while simultaneously satisfying acoustic properties. Vilchez et al. [33] proposed a simple and straightforward method to produce nonwovens with feather additions and (nano)cellulose fibres. The fabricated materials had good mechanical properties and seemed to meet the ecological criteria. The literature [28,42,43] reports that the use of feathers in nonwoven fabrics could find applications for erosion control purposes, especially in areas that have been denuded of vegetation and soil stabilisation is desired.

The developed nonwovens that could be used for agricultural applications are characterised by new functionalities, such as biodegradability that can be adjusted ad hoc to the type and duration of the crop and soil enrichment from the inflow of organic nitrogen. Nonwovens that only contain ingredients of natural origin are a valuable source of nutrients for plants. The flow of organic nitrogen into the soil comes from the biodegradation of the feathers.

## 5. Conclusions

The tested materials in both groups differed significantly. The positive effects of feather addition on the time of the biodegradation of the nonwovens in Group II were shown. Moreover, a correlation between the selected mechanical properties and the time of biodegradation was noticed. It could be concluded that the addition of feather waste affects the mechanical properties of the materials. The addition of feathers makes the product more susceptible to the action of microorganisms, releases C and N into the soil, shortens the biodegradation time and allows for waste management. The presented results of biodegradation and ecotoxicity confirmed the legitimacy of implementing this type of technology on a large scale. The biodegradable nonwovens that were tested and described in this paper seem to meet all criteria for eco-friendly agricultural products and have great potential for commercial use

## 6. Patents

P.430284. (19 June 2019) “Sposób wytwarzania puszystej włókniny kompozytowej”.

## Figures and Tables

**Figure 1 polymers-14-02370-f001:**
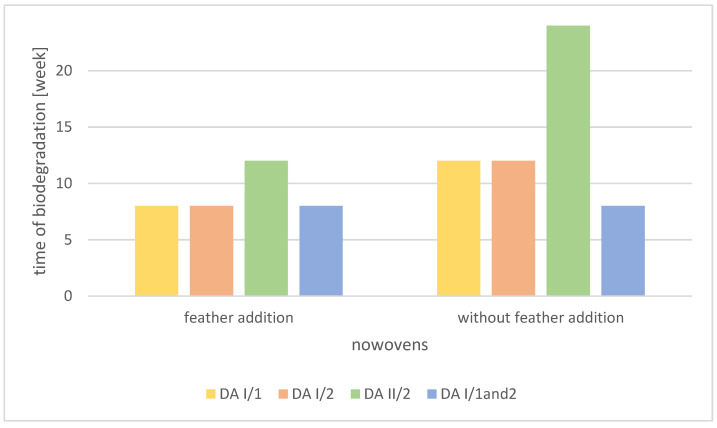
The biodegradation times of the samples, with and without the addition of feathers, for the samples in Group II.

**Figure 2 polymers-14-02370-f002:**
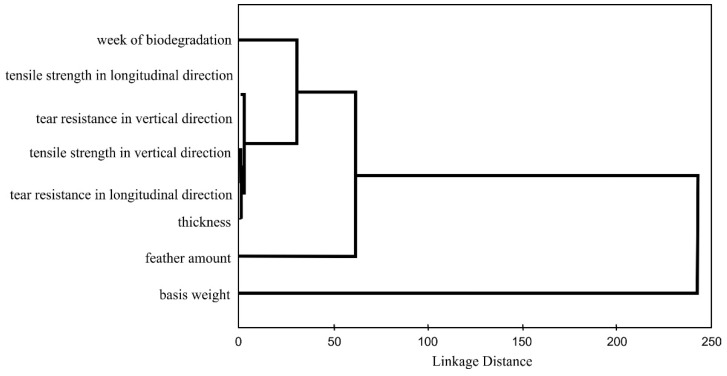
The biodegradation times of the samples, with and without the addition of feathers, in Group II.

**Table 1 polymers-14-02370-t001:** The compositions of the tested nonwovens.

Group	Nonwoven	Feather Amount (%)	Wool Amount (%)	Trevira Bico	
				Amount (%)	Type
I	Nonwoven I	0	90	10	256
		38.5	55	6	
	Nonwoven II	0	90	10	
		34.8	58	7	
	Nonwoven III	0	90	10	
		44.4	50	6	
II	DA I/1	30.0	63	7	453
		0	90	10	
	DA I/1 and 2	32.0	61	7	
		0	90	10	
	DA I/2	40	54	6	
		0	90	10	
	DA II/2	40.0	54	6	
		0	90	10	

**Table 2 polymers-14-02370-t002:** The mechanical properties of the tested materials (± SD).

Group	Nonwoven	Feather	Base Weight (g/m^2^)	Thickness (mm)	Tensile Strength in the Horizontal Direction (N)	Tensile Strength in the Vertical Direction (N)	Tear Resistance in the Horizontal Direction (N)	Tear Resistance in the Vertical Direction (N)
			EN 29073-1:1994	EN ISO 9073-2:2002	EN 29073-3:1994	EN 29073-3:1994	EN ISO 9073-4:2002	EN ISO 9073-4:2002
I	Wool I	+	78.2 ± 2.3	1.33 ± 0.06	1.11 ± 0.22	2.02 ± 0.43	1.80 ± 0.08	1.22 ± 0.04
		-	48.1 ± 4.5	1.13 ± 0.05	1.18 ± 0.60	2.68 ± 0.57	2.26 ± 0.30	1.45 ± 0.36
	Wool II	+	158.0 ± 7	2.04 ± 0.08	3.91 ± 1.07	10.20 ± 2.60	5.18 ± 0.39	4.94 ± 0.31
		-	103.0 ± 10.0	1.75 ± 0.08	4.89 ± 1.84	7.22 ± 1.09	5.89 ± 0.91	3.21 ± 0.35
	Wool III	+	284.0 ± 8.0	2.83 ± 0.18	31.9 ± 12.1	27.20 ± 2.20	31.2 ± 6.3	15.4 ± 2.2
		-	158.0 ± 12.0	2.11 ± 0.07	14.90 ± 2.40	40.00 ± 5.1	18.90 ± 3.20	23.50 ± 6.1
II	DA I/1	+	103.0 ± 6	1.62 ± 0.12	0.36 ± 0.11	0.70 ± 0.15	1.06 ± 0.56	0.52 ± 0.46
		-	86.0 ± 4.00	1.76 ± 0.13	0.58 ± 0.05	1.14 ± 0.27	1.43 ± 0.51	0.65 ± 0.12
	DA I/1 and 2	+	116.0 ± 13.0	1.70 ± 0.14	0.44 ± 0.05	0.92 ± 0.16	1.17 ± 0.36	0.38 ± 0.20
		-	70.9 ± 3.80	1.56 ± 0.12	0.56 ± 0.10	1.01 ± 0.21	1.02 ± 0.30	0.64 ± 0.57
	DA I/2	+	101.0 ± 14.0	1.75 ± 0.15	0.34 ± 0.07	1.44 ± 1.29	1.38 ± 0.04	0.43 ± 0.12
		-	86.0 ± 4.00	1.76 ± 0.13	0.58 ± 0.05	1.14 ± 0.27	1.43 ± 0.51	0.65 ± 0.12
	DA II/2	+	144 ± 0.16	2.14 ± 0.13	0.69 ± 0.14	1.70 ± 0.46	1.94 ± 0.32	0.85 ± 0.34
		-	90.8 ± 5.00	1.93 ± 0.15	0.72 ± 0.12	2.42 ± 0.51	2.74 ± 0.68	0.67 ± 0.14

**Table 3 polymers-14-02370-t003:** The photo-documentation of the biodegradation process (Sample DA II/2).

	DA II/2 “0”	DA II/2
1 Week	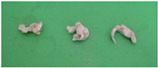	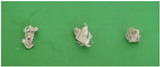
4 Weeks	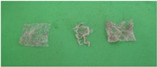	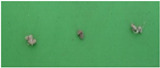
8 Weeks	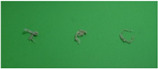	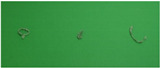
12 Weeks	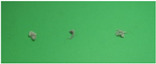	Biodegradation 100%
16 Weeks	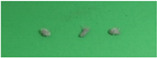	
20 Weeks	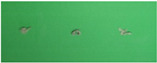	
24 Weeks	Biodegradation 100%	

**Table 4 polymers-14-02370-t004:** The results obtained by the ANOVA test for the samples in Group II.

		Feather	
		No	Yes
Week of Biodegradation	Mean	14	9
	SD *	6.93	2.00
	Min.	8	8
	Max.	24	12
	Median	12	8
Tensile Strength in the Horizontal Direction	Mean	0.61	0.46
	SD *	0.07	0.16
	Min.	0.56	0.34
	Max.	0.72	0.69
	Median	0.58	0.40
Tensile Strength in the Vertical Direction	Mean	1.42	1.19
	SD *	0.66	0.46
	Min.	1.01	0.70
	Max.	2.42	1.70
	Median	1.14	1.18
Tear Resistance in the Horizontal Direction	Mean	1.65	1.39
	SD *	0.75	0.39
	Min.	1.02	1.06
	Max.	2.74	1.94
	Median	1.43	1.28
Tear Resistance in the Vertical Direction	Mean	0.65	0.54
	SD *	0.01	0.21
	Min.	0.64	0.38
	Max.	0.67	0.85
	Median	0.65	0.47

SD *, standard deviation.

**Table 5 polymers-14-02370-t005:** The relationships between the metrological parameters and the biodegradability/biodegradation times of samples in Groups I and II (correlation matrix; correlation coefficient).

	Group	Feather Addition	Thickness	Tensile Strength in the Horizontal Direction	Tensile Strength in the Vertical Direction	Tear Resistance in the Horizontal Direction	Tear Resistance in the Vertical Direction
**Biodegradability (%)**	I	0.24	−0.16	0.12	−0.10	0.19	−0.05
**Weeks of Biodegradation**	II	**−0.62**	0.54	**0.74**	**0.87**	**0.94**	0.45

**bold**: values for which the hypothesis (H_0_; value of the coefficient in the correlation is 0) can be rejected.

## Data Availability

Not applicable.

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
