# Peer review of "Biodegradable Nonwovens with Poultry Feather Addition as a Method for Recycling and Waste Management"

_polymers, 2022, doi:10.3390/polym14122370_

Round 1

Reviewer 1 Report

Authors reported a very brief research on poultry feathers to waste textile exploring the biodegradative process of the mix.

Data should be provided together with their uncertainties. Otherwise, they are not scientific data but only numbers.

This is the real week point of this research that preclude the use of statistical tool for supporting the conclusions. I strongly recommend to deep revising this manuscript including experimental repetitions and statistical data elaboration with the appropriated tool (correlation factor, ANOVA test, significance test).

As it is presented now, I cannot endorse the publication of this research but the experimental approach is quite good and it could be considered after major revisions.

Author Response

Response to Reviewer 1 Comments

Point 1: Data should be provided together with their uncertainties. Otherwise, they are not scientific data but only numbers. This is the real week point of this research that preclude the use of statistical tool for supporting the conclusions. I strongly recommend to deep revising this manuscript including experimental repetitions and statistical data elaboration with the appropriated tool (correlation factor, ANOVA test, significance test).

 Response 1: Thank you for review. We agree with the reviewer's comment. The data has been supplemented with uncertainties. The statistical analysis has been revised and expanded. The obtained results were discussed.

Reviewer 2 Report

Comments to authors are below:

  • The abstract lacks to present the numerical values from significant findings in this paper.
  • Regarding the experimental section, authors should develop new works with some significant measurements to improve the quality of  the work presented in this paper.
  • Regarding the discussion of results, no new insight has been made, and limitation in the tests of this manuscript to make an impact based on the results presented.
  • The discussion of results is tremendously poor and  brief based on the limitation in the tests of the manuscript.

I do not find the results are presented enough and of great interest for publication in Polymers Journal

Author Response

Response to Reviewer 2 Comments

Point 1: The abstract lacks to present the numerical values from significant findings in this paper.

Response 1: Thank you for the comment. On the "abstract" section, the most important data has been added.

Point 2: Regarding the experimental section, authors should develop new works with some significant measurements to improve the quality of the work presented in this paper.

Response 2: Thank you for the comment. The suggestion will be taken into account in further research and scientific work. The presented results met the design assumptions. Due to the duration of the biodegradation tests, it is not possible to perform further tests in such a short time.

Point 3: Regarding the discussion of results, no new insight has been made, and limitation in the tests of this manuscript to make an impact based on the results presented. The discussion of results is tremendously poor and  brief based on the limitation in the tests of the manuscript.

Response 3: We agree with the reviewer's comment. The description of the methods has been extended (purpose, validation information, reference materials, time limits). Other tests used in the biodegradation assessment are discussed. The discussion of the results has been extended and refers to the results obtained.

Reviewer 3 Report

The article under consideration is devoted to the study of biodegradable nonwoven materials with the addition of bird feather. The positive effect of adding waste containing keratin on the process of biodegradation of the studied materials is shown. In addition, such an addition has been shown to be non-ecotoxic. The work was done at a high level, the conclusions correspond to the data obtained. It should be noted the high practical significance of this work and its commercial potential (the authors obtained a patent). I believe that the article can be published without changes.

Author Response

Response to Reviewer 3 Comments

Point 1: The article under consideration is devoted to the study of biodegradable nonwoven materials with the addition of bird feather. The positive effect of adding waste containing keratin on the process of biodegradation of the studied materials is shown. In addition, such an addition has been shown to be non-ecotoxic. The work was done at a high level, the conclusions correspond to the data obtained. It should be noted the high practical significance of this work and its commercial potential (the authors obtained a patent). I believe that the article can be published without changes.

Response 1: Thank you very much for your review.

Reviewer 4 Report

This paper, entitled Biodegradable nonwovens with poultry feather addition as a way of recycling and waste management , is a scholarly work and can increase knowledge on this domain. The authors provide an interesting and original work, the content is relevant to Polymers. The manuscript is quite well written and well related to exsting literature. The abstract and keywords are meaningful.

I have some specific and general comments:

  • Authors should mention in the introduction section that this application was patented if I understand right. Some details about partnership and development could be added in this section. Moreover the description of the main objectives of this study should be better introduced, in order to increase the quality of the whole manuscript.
  • About Materials and methods section, where is located the poultry slaughterhouse? What is the total amount of feathers that could be valorized by this application? This point should be discussed in the Discussion or Conclusion section, introducing economic consideration such as costs of operation, costs saving by this way vs conventional treatment of feathers,...
  • About Table 1, how were determined the ratio of each component for the composition of tested nowwovens? Is it from previous studies or assays, or from literature?
  • Please provide accuracy for data in Table 1. Same comment for Table 2.
  • Why Table 2 is in Materials and methods section, is it more results of characterization? Please discuss these values and parameters. Which parameter is the most important for the choice of materials? Is it possible to carry out a ranking or it's more a mix of parameters?
  • How were carried out biodegradability tests? How many samples and how many replication? What is the amount for each sample?
  • Mechanical properties were determined according to international standards as listed lines 143 to 145 page 5, please provide more details. What are these standards?
  • Please provide additional information for Methods such as methods and protocols used (references) for some of them, in particular mass loss.
  • Is there any material as control used for bidegradation "calibration" or "validation" of the test? Table 3.
  • Please insert legend below the Figures (Figure 1 and Figure 2).
  • The Figure 1 is not very easy to understand, please try to increase the readability.
  • Is there any other biodegradation tests that could be carried out instead of those selected in this study? If yes, please discuss about the advantages of this one. Please discuss gains or advantages vs limits.
  • What is the objective targeted in terms of biodegradation? Is there a limit of time or of size? Please discuss this.
  • The Result section is quite descriptive and should be improved in terms of interpretation and discussion of results.

As it, this manuscript is not fully acceptable for publication and requires some amendments and modification. Some points are missing and additional information are required. The interpretation and discussion of results must be improved also.I recommend the following decision: RECONSIDER AFTER MAJOR REVISION.

Author Response

Response to Reviewer 4 Comments

At the beginning, we would like to thank you for your very constructive opinion on the article.

Point 1: Authors should mention in the introduction section that this application was patented if I understand right. Some details about partnership and development could be added in this section. Moreover the description of the main objectives of this study should be better introduced, in order to increase the quality of the whole manuscript.

Response 1: We agree with the comment of the reviewer that it is beneficial to present the detailed information on patent and cooperation. patent (line 119), details about partnership (line 145-147), description of the objectives (148-158).

Point 2: About Materials and methods section, where is located the poultry slaughterhouse? What is the total amount of feathers that could be valorized by this application? This point should be discussed in the Discussion or Conclusion section, introducing economic consideration such as costs of operation, costs saving by this way vs conventional treatment of feathers,...

Response 2: Thank you for indicating the lack of important information. The data has been completed in the indicated section (line 145-147).

Point 3: About Table 1, how were determined the ratio of each component for the composition of tested nowwovens? Is it from previous studies or assays, or from literature?

Response 3: Thank you for your comment. The % content of feathers in the nonwoven was estimated on the basis weight study: the difference in grammage of the nonwoven containing the feathers and the grammage of the reference nonwoven (without feathers). The information was added in the text (line 149-152).

Point 4: Please provide accuracy for data in Table 1. Same comment for Table 2.

Response 4: Thank you for valuable comment. The data has been added to the Table 2.

Regarding Table 1: Due to the fact that the production of the nonwovens is carried out on a semi-technical scale, the presented composition of the samples is approximate.

Point 5: Why Table 2 is in Materials and methods section, is it more results of characterization? Please discuss these values and parameters. Which parameter is the most important for the choice of materials? Is it possible to carry out a ranking or it's more a mix of parameters?

Response 5: Thank you for the comment. Table 2 has been moved to the "results" section. The obtained values were subjected to additional tests (Table 4 and 5) and discussed (line 346-371)

Point 6: How were carried out biodegradability tests? How many samples and how many replication? What is the amount for each sample?

Response 6: Thank you for indicating the lack of important information. The data has been completed in the “materials and methods” section (line 162-168)

Point 7: Mechanical properties were determined according to international standards as listed lines 143 to 145 page 5, please provide more details. What are these standards?

Response 7: Thank you for the comment. Information on the scope of application of the standards has been added to Table 2.

Point 8: Please provide additional information for Methods such as methods and protocols used (references) for some of them, in particular mass loss. Is there any material as control used for bidegradation "calibration" or "validation" of the test? Table 3.

Response 8: Thank you for the comment. The data has been added in the “materials and methods” section (line 162-168).

Point 9: Please insert legend below the Figures (Figure 1 and Figure 2). The Figure 1 is not very easy to understand, please try to increase the readability.

Response 9: Thank you for indicating the lack of important information. For Figure 1, a legend has been added and clarity has been improved. However, in our opinion, adding a legend is impossible for Figure 2. If the reviewer has specific guidelines, please provide feedback.

Point 10: Is there any other biodegradation tests that could be carried out instead of those selected in this study? If yes, please discuss about the advantages of this one. Please discuss gains or advantages vs limits.

Response 10: Thank you for the comment. A description of alternative biodegradation methods has been added in the "introduction" section (line 43-52)

Point 11: What is the objective targeted in terms of biodegradation? Is there a limit of time or of size? Please discuss this.

Response 11: Thank you for the comment. The data has been added in the “materials and methods” section (line 162-168).

Point 12: The Result section is quite descriptive and should be improved in terms of interpretation and discussion of results.

Response 12: Thank you for the valuable comment. In response to the suggestion, the results and discussion section has been revised. The description of the obtained results has been extended.

Round 2

Reviewer 1 Report

Authors improved the manuscript as required. The paper is now matching the requirements for publications Accordingly, i endorse it.

Author Response

 Thank you very much for the review.

Reviewer 2 Report

Comments to author are listed below:

 Regarding the experimental section. Much information were missing and need more explanations.

Further experimental works should be performed to monitoring the biodegradations changes such as, using FTIR, thermal characterisations (DSC analysis) to improve the quality of this paper.

 Regarding the discussion of results, no new insight has been made, and limitation in the tests of this manuscript to make an impact based on the results presented.

The conclusions represent nothing in relation to the results obtained.

I do not find the results are presented enough and of great interest for publication in Polymers Journal.

Author Response

Response to Reviewer 2 Comments

Point 1: Regarding the experimental section. Much information were missing and need more explanations. Further experimental works should be performed to monitoring the biodegradations changes such as, using FTIR, thermal characterisations (DSC analysis) to improve the quality of this paper.

Response 1: The described results are part of the implemented project, which did not include FTIR and DSC analyses. We understand the commentary and the advisability of supplementing the research, however, as mentioned previously, biodegradation testing is a long-term process and it is not possible to repeat the research at the moment. We will take the hint into account in subsequent works and projects.

Point 2: Regarding the discussion of results, no new insight has been made, and limitation in the tests of this manuscript to make an impact based on the results presented. The conclusions represent nothing in relation to the results obtained.

Response 2: We understand your concern and have done our best to revise the paper and add the information suggested. It should be emphasized that the main purpose of the manuscript was to present the biodegradability of the produced nonwovens as a method of waste management. The obtained results were interpreted in line with comments from other Reviewers.

Reviewer 4 Report

The authors provide a revised version of their manuscript taking into account all the comments and requests of amendment made in the previous review. The authors provide detailed answers and amendments in the revised text. I agree with all the amendments and answers.

There's some small modification of form to carry out before publication.

- some paragraphs or lines are redundant (please see lines 164 to 170, there's three times the same paragraph lines 171 to 177 and 187 to 193), what is the right location of this paragraph?

- Why deleting text lines 249 to 258?

- Figure 2 is inserted twice times

- Sentence "Natural fibers are divided into three main groups, which arise from their origin: plant" is inserted five times (lines 310 to 314).

As it, but after these last modification, the manuscript could be acceptable for publication, before this last revision, I recommend the following decision: ACCEPT AFTER MINOR REVISION.

Author Response

Response to Reviewer 4 Comments

Point 1: some paragraphs or lines are redundant (please see lines 164 to 170, there's three times the same paragraph lines 171 to 177 and 187 to 193), what is the right location of this paragraph?

Response 2: Sorry for our inattention. The the repeat has been deleted.

Point 2: Why deleting text lines 249 to 258?

Response 2: The text has been moved (lines 312-318)

Point 3: Figure 2 is inserted twice times. Sentence "Natural fibers are divided into three main groups, which arise from their origin: plant" is inserted five times (lines 310 to 314).

Response 3: Thank you for the comment. Repeats may be due to technical problems with loading the file. Both in the version loaded and downloaded from the system, such repetitions do not occur.
